# Anti-Cancer, Anti-Angiogenic, and Anti-Atherogenic Potential of Key Phenolic Compounds from Virgin Olive Oil

**DOI:** 10.3390/nu16091283

**Published:** 2024-04-25

**Authors:** Ana Dácil Marrero, Ana R. Quesada, Beatriz Martínez-Poveda, Miguel Ángel Medina

**Affiliations:** 1Facultad de Ciencias, Departamento de Biología Molecular y Bioquímica, Andalucía Tech, Universidad de Málaga, E-29071 Málaga, Spain; anadacil@uma.es (A.D.M.); quesada@uma.es (A.R.Q.); bmpoveda@uma.es (B.M.-P.); 2Instituto de Investigación Biomédica y Plataforma en Nanomedicina-IBIMA Plataforma BIONAND (Biomedical Research Institute of Málaga), E-29071 Málaga, Spain; 3CIBER de Enfermedades Raras (CIBERER), Instituto de Salud Carlos III, E-28029 Madrid, Spain; 4CIBER de Enfermedades Cardiovasculares (CIBERCV), Instituto de Salud Carlos III, E-28029 Madrid, Spain

**Keywords:** Mediterranean diet, chemoprevention, hydroxytyrosol, oleacein, oleocanthal, hydroxytyrosol derivatives

## Abstract

The Mediterranean diet, renowned for its health benefits, especially in reducing cardiovascular risks and protecting against diseases like diabetes and cancer, emphasizes virgin olive oil as a key contributor to these advantages. Despite being a minor fraction, the phenolic compounds in olive oil significantly contribute to its bioactive effects. This review examines the bioactive properties of hydroxytyrosol and related molecules, including naturally occurring compounds (-)-oleocanthal and (-)–oleacein, as well as semisynthetic derivatives like hydroxytyrosyl esters and alkyl ethers. (-)-Oleocanthal and (-)–oleacein show promising anti-tumor and anti-inflammatory properties, which are particularly underexplored in the case of (-)–oleacein. Additionally, hydroxytyrosyl esters exhibit similar effectiveness to hydroxytyrosol, while certain alkyl ethers surpass their precursor’s properties. Remarkably, the emerging research field of the effects of phenolic molecules related to virgin olive oil on cell autophagy presents significant opportunities for underscoring the anti-cancer and neuroprotective properties of these molecules. Furthermore, promising clinical data from studies on hydroxytyrosol, (-)–oleacein, and (-)–oleocanthal urge further investigation and support the initiation of clinical trials with semisynthetic hydroxytyrosol derivatives. This review provides valuable insights into the potential applications of olive oil-derived phenolics in preventing and managing diseases associated with cancer, angiogenesis, and atherosclerosis.

## 1. Mediterranean Diet: More Than a Dietary Choice

Derived from the culinary traditions of the Mediterranean region, the Mediterranean diet has captured significant attention in both academic discourse and public health research. This is primarily due to its well-documented association with numerous health benefits. Centered around the predominant consumption of an array of healthy food products, this dietary pattern is not only renowned for its appetizing flavors but also for its potential to mitigate the risk of chronic diseases, extend life expectancy, and enhance overall well-being [1,2]. With centuries of historical heritage and a growing body of empirical evidence, the Mediterranean diet stands as an exemplary case study for investigating the intricate relationship between culture, nutrition, and health [3,4].

Since the Mediterranean diet derives from the traditional way of eating in countries bordering the Mediterranean Sea, such as Spain, Italy, Greece, Morocco, Tunisia, Libya, and others, there is not just one Mediterranean diet; rather, it varies slightly from one country to another, adapting to their unique cultures and lifestyles. However, the use of virgin olive oil (VOO) is a common feature in all these dietary variations [1,4]. Notably, UNESCO has recognized the Mediterranean diet as an intangible cultural heritage deeply connected to its place of origin, promoting responsible interactions with the environment through its agricultural and culinary practices [5].

The Mediterranean diet typically features a high intake of olive oil, olives, fruits, vegetables, unrefined cereals, legumes, and nuts (Figure 1). In addition, it includes a moderate consumption of fish and dairy products, while meat products are consumed in lower quantities [4]. One controversial aspect of the Mediterranean diet is the inclusion of red wine, which contains a higher concentration of phytochemical compounds compared to other alcoholic beverages. An ongoing debate centers around whether the potential benefits of bioactive molecules in red wine outweigh the potential drawbacks of alcohol consumption, even when consumed moderately [6]. Furthermore, considerations such as religious and social factors that may limit alcohol intake should be considered [1,7].

A wide array of health benefits has been associated with the Mediterranean diet. The most consistent and robust body of evidence supports its positive impact on cardiovascular risk factors and cardiovascular disease [8,9,10,11,12,13]. Furthermore, a substantial body of literature also highlights its potential benefits for many other health outcomes, including diabetes [14,15,16,17], obesity [18,19], cancer [20,21,22,23,24], and cognitive decline [25,26], among others [8,13]. 

Despite all the reported benefits to health, measuring and implementing a complete Mediterranean diet presents challenges, resulting in limited interventional studies conducted on humans. The majority of these studies are of short-term duration and focus on the measurement of intermediate factors, such as plasma lipid concentrations, inflammation markers, blood pressure, fasting glucose, and weight loss, rather than assessing the ultimate disease outcomes like myocardial infarction, stroke, dementia, and cancer development [1]. Contrary to this, the PREDIMED study aimed to establish a strong correlation between the Mediterranean diet and a reduced incidence of cardiovascular events [9]. Similarly, the CORDIOPREV study, which has been completed but is awaiting results, seeks to assess the impact of the Mediterranean diet compared to a low-fat diet in the secondary prevention of cardiovascular diseases [27].

An advantage of the Mediterranean dietary pattern lies in the concept that “the whole is greater than the sum of its parts”. In other words, the protective effect of consuming the typical Mediterranean diet foods is more significant when they are consumed together. Notably, the consumption of VOO should be emphasized, as it serves as a common thread in all variations of the Mediterranean diet across different regions and countries. An overview of the bioactivities of VOO natural-contained and synthetic-derived phenolic compounds regarding anti-cancer, anti-angiogenic, and anti-atherogenic effects is the main focus of this review.

## 2. Virgin Olive Oil

Olive oil is a product derived from the mechanical extraction of the fruit *Olea europaea* L. (*Oleaceae* family) [28]. Among the various types of olive oils, VOO stands out for its method of production. It is obtained through mechanical or other physical processes, such as washing, decantation, centrifugation, or filtration, conducted under specific thermal conditions that do not cause any alterations in the oil [29]. Importantly, these high-quality oils, namely extra virgin olive oil (EVOO) and VOO, are never subjected to solvent-based extraction, chemical treatments, re-esterification processes, or any mixing with other types of oils [29,30].

The composition of olive oil is predominantly lipidic, with triacylglycerols accounting for approximately 99% of its composition. Additionally, it contains smaller proportions of free fatty acids, mono- and diacylglycerols, and various lipids, including hydrocarbons, sterols, aliphatic alcohols, tocopherols, and pigments. Triacylglycerols in VOO are rich in monounsaturated fatty acids (MUFAs), particularly oleic acid, with palmitic acid as the main saturated fatty acid, and some polyunsaturated fatty acids (PUFAs), such as linoleic and linolenic acid [28,29,30,31] (Table 1). 

Traditionally, the cardioprotective effects of VOO have been attributed to its lipid profile, which has indeed been linked to improvements in cardiovascular parameters like low-density lipoprotein (LDL) levels [34,35,36]. However, it is now recognized that many of these benefits, among others, are primarily mediated by minor components of this food, namely, phenolic compounds, which play a pivotal role in determining the biological properties and sensory attributes of VOO [29,31,37].

Together with molecules like carotenoids and tocopherols, phenolic compounds are the primary antioxidants found in VOO. These compounds play a crucial role in determining the oil’s sensory attributes, such as bitterness, pungency, and stability, shaping the organoleptic characteristics of aroma and flavor in each VOO [28,29]. However, what is particularly intriguing is that these antioxidants not only enhance the organoleptic properties and stability of VOO but also serve vital roles in human health, exhibiting antioxidant properties [38], among other beneficial effects.

## 3. Bioactive Phenolic Compounds in Virgin Olive Oil and Their Derivatives

The concentration of phenolic compounds in VOO exhibits significant variation due to agronomic factors, including the geographical origin, olive cultivar, fruit ripeness stage, and extraction process. This concentration can span from 50 to 940 mg/kg of oil [32,33], with concentrations typically falling within the range of 100 to 300 mg/kg [29], illustrating substantial differences among different varieties and conditions.

In terms of their chemical structure, phenolic compounds found in VOO encompass a diverse array of molecules from various classes (Table 2). These include simple and alcoholic phenols, such as vanillic, gallic, coumaric, and caffeic acids, as well as tyrosol and hydroxytyrosol. Additionally, VOO contains more complex compounds like secoiridoids (e.g., oleuropein and ligstroside, along with their derivatives), lignans, hydroxychromans, and flavones. Notably, among these compounds, alcoholic phenols and secoiridoids are present in substantial quantities in VOO [28,39].

Remarkably, the alcoholic phenols, tyrosol and hydroxytyrosol, and the secoiridoids share close structural relationships (Figure 2). Tyrosol (*p*-hydroxyphenethyl ethanol, *p*-HPEA (**1**)) and hydroxytyrosol (3,4-dihydroxyphenylethyl ethanol, 3,4-DHPEA (**2**)) can exist independently in VOO. However, upon esterification with elenolic acid, they undergo derivatization into secoiridoids. These compounds constitute a prominent and intricate family within the polar fraction of VOO. The most prevalent secoiridoids in VOO include the mono-aldehydic forms of ligstroside aglycones (*p*-HPEA-EA (**3**)) and oleuropein (3,4-DHPEA-EA (**4**)), as well as the di-aldehydic forms of their decarboxymethylated derivatives, (-)–oleocanthal (*p*-HPEA-EDA (**5**)) and (-)–oleacein (3,4-DHPEA-EDA (**6**)) [40]. 

All the previously mentioned molecules occur naturally in VOO and exert remarkable bioactivities that will be further discussed. Interestingly, in the contemporary research landscape, a considerable effort is dedicated to exploring novel formulations of hydroxytyrosol to enhance its absorption, distribution, metabolism, and excretion (ADME) processes, as well as improving its stability and biological health features. These investigations primarily concentrate on altering the solubility of hydroxytyrosol to enhance both its bioavailability and plasma half-life [41]. A comprehensive review conducted by Bernini and colleagues examines various newly isolated hydroxytyrosol-derived compounds, their associated biological activities, and synthesis methods. Noteworthy categories of these compounds include hydroxytyrosol esters, hydroxytyrosol alkyl ethers, and others such as hydroxytyrosol analogs, thioderivatives and hydroxytyrosol-derived isochromans [42].

Hydroxytyrosol esters are some of the most studied synthetic derivatives of hydroxytyrosol. Interestingly, once hydroxytyrosol esters enter a cell, they can be hydrolyzed by cell lipases and esterases, thus possibly generating both the precursor hydroxytyrosol, which is largely responsible for the observed bioactivities, and a novel molecule that might be responsible for different activities [42]. Hydroxytyrosol acetate is one of the most prominent molecules within this group and can also be naturally present in VOO. Furthermore, representing a novel class of lipophilic hydroxytyrosol derivatives, hydroxytyrosyl alkyl-ethers exhibit improved chemical stability compared to hydroxytyrosol and hydroxytyrosol acetate [43]. Contrary to hydroxytyrosol esters, these compounds will likely remain invariant once inside the cell and act in either a similar or a different way compared to their precursor [42]. The molecular structures of hydroxytyrosol esters and alkyl ethers are provided in Figure 3.

Numerous studies have highlighted the positive impacts of phenolic compounds derived from VOO on human health. However, while tyrosol and hydroxytyrosol have been extensively studied, other structurally related molecules, such as (-)–oleocanthal and, especially, (-)–oleacein, have garnered comparatively less attention. Furthermore, while not naturally occurring in VOO, the semisynthetic derivatives of hydroxytyrosol deserve recognition due to their promising outcomes in enhancing the effects of their precursors and the growing interest in their development. This review focuses on key pre-clinical and clinical investigations into the antioxidant, anti-cancer, anti-angiogenic, and anti-atherosclerotic effects of hydroxytyrosol and some of its derivatives and related compounds.

## 4. Preclinical Data on the Anti-Cancer, Anti-Angiogenic, and Anti-Atherosclerotic Effects of Primary Phenolic Compounds from Virgin Olive Oils

### 4.1. Antioxidant Characteristics 

Despite having been extensively investigated, we deemed it crucial to incorporate a concise discussion on the primary mechanisms governing the antioxidant properties of key phenolic molecules from VOO. This inclusion is considered significant, given the profound interconnection between these antioxidant attributes and various anti-cancer and anti-atherosclerotic properties, which will be further addressed. In this section, we examine the antioxidant properties exerted by key phenolic compounds found in VOOs (Table 3), mainly reactive oxygen species (ROS) scavenging or the promotion of antioxidant systems, which encompasses their roles in cellular protection mechanisms. 

Among tyrosol and hydroxytyrosol, the latter exerts the most important health-related effects. Tyrosol exhibits antioxidant capacities, although they are generally weaker than hydroxytyrosol. Nonetheless, tyrosol remains an effective cellular antioxidant, likely due to its intracellular accumulation [64]. It is worth noting that tyrosol can be converted into hydroxytyrosol in the liver through the enzymatic activity of cytochrome P450 (CYP) CYP2A6 and CYP2D6, suggesting that tyrosol may serve as a precursor to hydroxytyrosol [65,66]. Regarding its biological effects, antioxidant [67,68] and anti-inflammatory properties [69] have been described for this molecule. However, in comparative studies, hydroxytyrosol typically exhibits stronger biological properties than tyrosol, as observed in various contexts, including antioxidant [70], anti-angiogenic [71], and anti-atherogenic effects [72].

The antioxidant properties of hydroxytyrosol arise from its capacity to function as a potent scavenger of free radicals [44,45,46], its chelating effect on metals—diminishing the production of ROS from derived reactions [47]—and its ability to stimulate various antioxidant systems by promoting the activity of enzymes like catalase (CAT), superoxide dismutase (SOD), and glutathione peroxidase (GPx) [48,49,50,51], or by activating the transcription factor Nrf2 [52,53]. Additionally, (-)–oleocanthal and (-)–oleacein, while not as well studied as hydroxytyrosol for their antioxidant properties, have been shown to act as free radical scavengers [54,55,56].

Hydroxytyrosol derivatives also yield remarkable antioxidant effects. Hydroxytyrosol acetate is one of the most prominent molecules within this group and is naturally occurring in VOO. Remarkably, hydroxytyrosol acetate and propionate exerted similar or slightly higher antioxidant capacities than hydroxytyrosol [57,58,60]. Nitro-ester derivatives have also been found to enhance the antioxidant properties of hydroxytyrosol [59]. Interestingly, in those studies where a set of derivatives with increasing lengths of chains were compared, a decrease in activity with longer side chains in keeping with their lipophilic nature was detected. Hydroxytyrosyl alkyl-ethers demonstrated robust antioxidant effects, surpassing those of hydroxytyrosol [61,62,73].

### 4.2. Anti-Cancer Properties

The anti-cancer properties of primary phenolic compounds extracted from VOOs stand as a crucial area of investigation in pharmacological research. Globally, cancer remains a predominant cause of mortality worldwide, contributing to nearly ten million deaths, or almost one in six deaths, in 2020 [74]. Despite substantial advancements in understanding the nature of cancer, the persistently high incidence of this disease underscores the necessity for innovative approaches that complement traditional interventions. Over recent decades, attention has been turned towards cancer chemoprevention, a concept initially enunciated by Sporn in 1976 and encompassing the use of natural, synthetic, or biological agents to reverse, suppress, or prevent tumor progression [75]. Moreover, numerous cancer types exhibit close associations with dietary patterns linked to the Western lifestyle, such as low fruit and vegetable consumption [76]. Therefore, diet becomes an invaluable tool in the focus on cancer chemoprevention. This section aims to elucidate the anti-tumoral attributes of important bioactive constituents of VOO, shedding light on their mechanisms and implications in mitigating cancer-related risks (Table 4).

Hydroxytyrosol exerts significant effects on cancer progression, which are worth highlighting. Notably, ROS play a pivotal role in tumor formation, contributing to cancer initiation, promotion, and progression. The anti-tumor properties of this phenolic compound were primarily attributed to its role as an ROS scavenger and its capacity to modulate the antioxidant system. However, over the past decade, numerous studies have focused on demonstrating exclusive anti-tumor effects [41,83]. In this line, hydroxytyrosol has been shown to inhibit tumor cell proliferation by inducing cell cycle arrest through the modulation of cyclins [77,80,81,82,83,84,85,86,87,88]. Additional mechanisms of impeding cell proliferation exerted by this compound include the inhibition of the extracellular signal-regulated kinase 1/2 (ERK1/2) pathway [77,78], the induction of apoptosis by activating caspases and the mitochondrial pathway [80,83,85,86,87], and the reduction in the pro-survival protein kinase B (AKT) signaling pathway [88]. Additionally, hydroxytyrosol has demonstrated other effects, such as reducing epidermal growth factor receptor (EGFR) levels in colon cancer cells [79] and affecting the WNT pathway in breast cancer models [80]. Interestingly, the molecule exhibited pro-oxidant effects specifically in cancer cells [89,90,91]. 

Clear anti-cancer effects have been documented for (-)–oleocanthal in various human cancers, including colorectal [92], prostate [96], breast [95,96], myeloma [94], and melanoma [97] and non-melanoma skin cancer [93], both *in vitro* and *in vivo*. These effects encompass defects in cell survival, proliferation, migration, and invasion, with underlying mechanisms including the reduction in hepatocyte growth factor (HGF) [95,96] and inhibition of the mammalian target of rapamycin (mTOR), STAT3 [97], and the AKT and ERK signaling pathways [92,93,94]. The induction of apoptosis in cancer cells by (-)–oleocanthal through the intrinsic or mitochondrial pathway [92,94] and decreased proliferation caused by cell cycle arrest [95] have also been observed. Furthermore, (-)–oleacein reduced the activation of the AKT and ERK signaling pathways, with implications for cancer and angiogenesis [93].

There is limited research on the anti-cancer effects of semisynthetic derivatives of hydroxytyrosol. However, studies have indicated a decrease in tumor cell proliferation through the inhibition of the AKT and ERK signaling pathways and cell cycle arrest, as well as a reduction in cell migration [58,98,99,101].

### 4.3. Modulatory Effects on Angiogenesis 

Angiogenesis, the process by which blood vessels originate from preexisting ones, is directly related to cancer progression [102,103,104]. In this context, the concept of angioprevention arises, denoting the prevention of cancer through the inhibition and/or stabilization of tumor angiogenesis [105]. Aligned with this notion, plant-derived compounds emerge as excellent angiopreventive candidates as their consumption entails low or non-existent toxicities, and they are readily accessible as integral components of dietary plant foods. Moreover, these compounds usually exhibit pleiotropic biological activity, extending their impact beyond tumor cells to include endothelial and immune cells. Importantly, other components of VOO, distinct from those addressed herein, have been suggested as angio-preventive phytochemicals [106]. Within this section, we comprehensively examine the existing literature concerning the modulatory mechanisms of hydroxytyrosol and its derivatives, (-)–oleocanthal, and (-)–oleacein on angiogenesis (Table 5) as modulators of angiogenesis and as angiopreventive molecules.

Both cancer-related and cancer-independent effects of hydroxytyrosol on angiogenesis have been documented. Hydroxytyrosol can inhibit cyclooxygenase 2 (COX-2) activity [107]. In addition, data support a reduction in the AKT [88,108], nuclear factor-kappa B (NF-κB) [88], and vascular endothelial growth factor receptor (VEGFR) signaling pathways [108] by this molecule. Remarkably, our research group demonstrated that hydroxytyrosol exhibited anti-angiogenic effects by targeting extracellular remodeling and reducing matrix metalloproteinases’ (MMPs) production [71,109], in alignment with the findings of Scoditti and colleagues [107].

Regarding the derivatives of secoiridoids, a modulatory effect could be foreseen provided the reduction in the CD31 microvessel marker when endothelial cells were treated with (-)–oleocanthal [96]. Remarkably, other groups, including ourselves, demonstrated this anti-angiogenic effect of (-)–oleocanthal directly on endothelial cells [97,111]. Anti-angiogenic effects of (-)–oleacein were suggested by Carpi and colleagues, involving the reduced expression of *VEGF*, *COX2* and *MMP2* [55]. In this line, our group has recently outlined potent anti-angiogenic properties of (-)–oleacein in vitro and in vivo [111].

Limited but promising findings underscore the potential angiogenesis-modulating properties of hydroxytyrosol derivatives. Research by our group has evidenced more robust anti-angiogenic activity for hydroxytyrosol acetate compared to its precursor [112]. Nonetheless, no enhanced effects on angiogenesis in vitro or in vivo were observed in this study, suggesting that the modulatory effects of these derivatives on angiogenesis are independent of their antioxidant capacity [112]. Notably, our group also documented enhanced angiogenesis-inhibiting effects of ethyl hydroxytyrosyl ether compared to hydroxytyrosol [112]. These results laid the foundation for a comparative study of a set of alkyl hydroxytyrosol ethers, among which hexyl hydroxytyrosyl ether (HT-C6) emerged as the derivative that showed the best anti-angiogenic performance [110].

### 4.4. Anti-Atherosclerotic Properties 

The investigation into the potential health benefits of dietary components has garnered significant attention in biomedical research, particularly in the context of preventing cardiovascular diseases. Atherosclerosis, characterized by the accumulation of lipid-rich plaque in arterial walls, stands as a major contributor to cardiovascular morbidity and mortality [113]. Historically considered a disease of cholesterol accumulation, it is currently recognized as an inflammation-driven syndrome [114]. In this context, the vascular endothelium actively contributes to the development of atherosclerosis through endothelial dysfunction. When exposed to various harmful stimuli, the endothelium undergoes a phenotypical shift towards such a maladaptive state, characterized by a specific gene expression profile that favors the expression of immune chemoattractant and adhesion molecules, thereby contributing to the progression of the pathology [115]. This section delves into an exploration of the molecular mechanisms governing the anti-atherosclerotic attributes inherent in primary phenolic compounds, with a focus on their hypolipidemic action and inflammation-related effects (Table 6). 

The anti-atherosclerotic and cardioprotective effects of hydroxytyrosol can also be related, or not, to its antioxidant properties. For instance, the antioxidant action of hydroxytyrosol yields protection of LDL from oxidation [45,49,50]. In addition, this phenolic compound promotes hypocholesterolemia itself, lowering plasma levels of cholesterol, LDL, and triglycerides and increasing high-density lipoprotein (HDL) levels [49,50]. Other effects rely on its anti-inflammatory capacity. As an example, hydroxytyrosol reduces the expression of proinflammatory cytokines like tumor necrosis factor-alpha (TNF-α) [51,117] and chemokines like C-C motif chemokine ligand 2 (CCL2) or C-X-C motif chemokine ligand 10 (CXCL10) [117]. In addition, it decreases the expression of adhesion molecules by the endothelium [72,120,121,122]. Furthermore, it reduces COX-2 activity [116,118] and decreases inducible nitric oxide synthase (iNOS) activity and nitric oxide (NO) production [44,116,117,118,119]. The mechanism behind hydroxytyrosol’s anti-inflammatory effects seems to be interfering with NF-κB signaling [51,72,116,117]. 

(-)–Oleocanthal has shown an inhibition of COX-2 [55,123,124], iNOS [123,125], and NF-κB [55,123] and its target genes. Furthermore, –(-)oleocanthal acts as an mTOR inhibitor [132]. Additionally, (-)–oleacein has been described to inhibit COX-2 [55,126,127] and NO production [127] and to reduce NF-κB signaling [55,56,126], resulting in decreased expression of proinflammatory cytokines and adhesion molecules in the endothelium.

Scarce preclinical studies address the anti-atherosclerotic effects of the derivatives of hydroxytyrosol. Nevertheless, the anti-inflammatory properties of hydroxytyrosol esters include COX-2 inhibition and the reduction in NO, proinflammatory cytokines’, and prostaglandins’ production [127,128,129]. Remarkably, hydroxytyrosyl alkyl-ethers demonstrated robust anti-inflammatory effects [63,130,131], surpassing those of hydroxytyrosol.

### 4.5. Effects on Autophagy

Markedly, a general overview of the impact of phenolic compounds derived from VOO on cell autophagy is worth acknowledging [133]. This influence contributes significantly to the overall beneficial effects of VOO polyphenols, particularly in addressing neurodegenerative conditions characterized by heightened oxidative stress and disruptions in proteostasis: the intricate process of eliminating protein accumulations and defective organelles [134]. 

Autophagy maintains cellular homeostasis through the removal and recycling of damaged macromolecules and organelles. In neurological contexts, phenolic compounds present in VOO, such as (-)–oleocanthal, have been documented to enhance autophagy through mTOR inhibition [132,135]. However, conflicting evidence from some studies suggests an inhibitory effect on autophagy [136,137]. 

Interestingly, the modulatory effect of autophagy by phenolic compounds has also been related to cancer progression. For instance, there appear to be autophagy-mediated anti-migration and invasive effects in tumor cell lines induced by hydroxytyrosol and oleuropein [138]. Nevertheless, the role of autophagy in cancer is also controversial and varies according to tumor type, stage, and therapy. Additionally, (-)–oleacein has emerged as a promising inhibitor of the histone demethylase LSD1/KDM1A. This enzyme holds a central epigenetic role in nutrient-driven metabolic adaptation and reprogramming, influencing multifactorial diseases such as obesity-associated disorders, neurological conditions, and cancer [139]. 

The evolving body of evidence positions the study of the effects of phenolic compounds on cell autophagy as a rapidly growing and intriguing field. Remarkably, its potential implications span diverse fields, including nutrition, and lead to the exploration of new therapeutic avenues for disorders such as cancer and neurodegenerative diseases.

## 5. Clinical Evidence on the Bioactivities of Key Phenolic Molecules from Virgin Olive Oil

A substantial body of clinical and correlational studies has addressed the positive impacts associated with VOO consumption in the realms of cancer and cardiovascular risk [9,13,22,23,26]. Despite the abundance of research in this field, limited attention has been directed towards the examination of individual components within VOO. This section provides a comprehensive overview of the existing clinical evidence, shedding light on the bioactivities of the principal phenolic constituents found in VOO and their relevance in the context of health promotion and disease prevention.

In the context of cancer research, a limited yet promising body of studies has emerged. One pilot study focused on assessing the impact of hydroxytyrosol on mammographic density among women at a heightened risk of developing breast cancer [140]. Notably, hydroxytyrosol demonstrated a significant reduction in breast density, particularly in women aged over 60 and those with elevated baseline breast density. Moreover, supplementation with hydroxytyrosol exhibited effects on tumor cell proliferation and influenced the WNT signaling pathway. While these findings are currently awaiting peer review, they offer a preliminary basis for future, more extensive investigations into the potential chemopreventive role of this natural compound in breast cancer. Additionally, an ongoing randomized clinical trial aims to elucidate the impact of hydroxytyrosol’s antioxidant and anti-inflammatory properties on the intestinal microbiota in patients with colon cancer, although conclusive results have yet to be reported [141]. 

As discussed previously, (-)–oleocanthal and (-)–oleacein exhibit significant in vitro and in vivo anti-tumor properties. However, data regarding their anti-cancer activity in humans are currently lacking. A pilot clinical study sought to determine the feasibility and tolerability of an intervention with VOO high in (-)–oleocanthal and (-)–oleacein for patients at an early stage of chronic lymphocytic leukemia (CLL) [142]. Remarkably, only VOO with elevated (-)–oleocanthal and (-)–oleacein content demonstrated beneficial effects on hematological and apoptotic markers. The intervention led to an increase in apoptotic markers, caspase-cleaved keratin 18 (CCK18) and Fas, and negative regulators of the cell cycle (p21), along with a decrease in the anti-apoptotic protein survivin and the cell cycle regulator cyclin D. These promising results suggest that this specific type of VOO could serve as a potential dietary intervention to enhance CLL outcomes by inducing cancer cell apoptosis and improving patient metabolism.

In the realm of atherosclerosis and the prevention of cardiovascular diseases, an increasing number of studies have delved into hydroxytyrosol and its related molecules. This exploration may be connected to the historically examined and discussed beneficial effects of the Mediterranean diet and VOO on cardiovascular health. In this line, a randomized, crossover trial added valuable insights, revealing that oral supplementation with hydroxytyrosol and punicalagin improved early atherosclerosis markers in middle-aged, healthy adults [143]. Notably, the supplement exerted anti-atherosclerotic effects by enhancing endothelial function, reducing blood pressure, and lowering circulating oxidized LDL levels, particularly in individuals with altered parameters. Similarly, a double-blind, controlled trial investigated the impact of the daily intake of VOO rich in various phenolic compounds on platelet reactivity in healthy adult males [144]. The study found that the only olive oil lacking beneficial effects on platelet aggregation was the one without (-)–oleacein and (-)–oleocanthal. Additionally, another study demonstrated that treatment with VOO rich in (-)–oleocanthal and (-)–oleacein substantially improved the oxidative and inflammatory status in people with obesity and prediabetes [145], in line with the results obtained in preclinical studies [146].

Moreover, several ongoing trials aim to further our understanding. For instance, one study evaluates the effects of a commercially available standardized olive extract (Tensiofytol^®^; Tilman, Baillonville, Belgium) containing 100 mg of oleuropein and 20 mg of hydroxytyrosol and administered daily to individuals with elevated blood pressure [147]. Another study explores the impact of chronic consumption of a hydroxytyrosol-rich extract from olives (Hytolive^®^; Genosa I+D, Málaga, Spain) on a population at high risk of age-related pathologies like type 2 diabetes and cardiovascular diseases [148]. Markers measured will include oxidative stress, inflammation, and glucose and lipid profiles. Lastly, an additional study compares two olive oil extracts, one containing hydroxytyrosol (Olivomed^®^; Intermed, Attica, Greece) and another containing a combination of oleuropein, hydroxytyrosol, and (-)–oleocanthal, examining their effects on endothelial, cardiac, and vascular functions in patients with coronary artery disease [149].

Shifting to the context of neurodegeneration and cognitive decline, few clinical studies are available. Nevertheless, a pilot study demonstrated that VOO consumption led to improvements in the blood–brain barrier function, enhanced brain function, and memory in individuals with mild cognitive impairments [150]. Importantly, findings revealed similar beneficial effects with refined olive oil (ROO), suggesting that the monounsaturated fats present in both VOO and ROO, such as oleic acid, contribute to these positive outcomes.

## 6. Concluding Remarks

The Mediterranean diet has gained significant recognition for its association with numerous health benefits. This dietary pattern, characterized by the consumption of healthy foods, not only offers flavorful meals but also demonstrates potential in mitigating the risk of chronic diseases, extending life expectancy, and enhancing overall well-being. 

Recent research focuses on VOO, a cornerstone of the Mediterranean diet, as a key contributor to this diet’s health-promoting properties. Currently, emphasis has shifted to the minor components of VOO, particularly phenolic compounds, rather than the lipidic fraction, recognizing them as the primary contributors to its biological effects. In this line, hydroxytyrosol and structurally related molecules, such as (-)–oleocanthal and (-)–oleacein, stand out due to their antioxidant, anti-inflammatory, anti-angiogenic, and anti-cancer properties. In addition, the role of phenolic molecules in VOO on autophagy is an emerging field with significant potential for understanding their mechanisms of action and identifying new therapeutic targets. 

Exploring the therapeutic potential of hydroxytyrosol derivatives, such as esters and alkyl ethers, presents a promising avenue for enhancing the bioactivities of the precursor or even finding new ones. Importantly, in the search and development of improved derivatives of this phenolic compound, the length and volume of the substituents seem crucial in determining optimal performance.

In clinical research, limited yet encouraging studies endorse the pharmacological promise of hydroxytyrosol, (-)–oleocanthal, and (-)–oleacein. Further investigations involving these compounds, either individually or in combination, as well as olive oils enriched with them, would enhance our understanding of their actual capacity for disease prevention and establish the foundation for clinical interventions. Moreover, certain hydroxytyrosol derivatives, particularly alkyl ethers, exhibit therapeutic potential that demands clinical evaluation.

In summary, the Mediterranean diet, with a central focus on VOO and its phenolic compounds, represents a holistic approach to health, offering a multitude of benefits that extend beyond cardioprotective effects and contribute to the prevention of prevalent diseases. The ongoing exploration of derivatives of these natural molecules and the expansion of our understanding of their effects on autophagy open new avenues for future research and therapeutic applications in various health issues.

## Figures and Tables

**Figure 1 nutrients-16-01283-f001:**
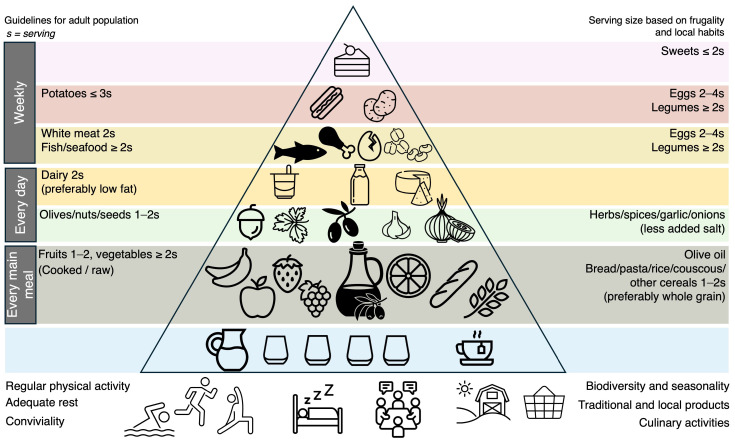
The Mediterranean diet pyramid. Inspired by “Fundación Dieta Mediterránea, 2010 edition”.

**Figure 2 nutrients-16-01283-f002:**
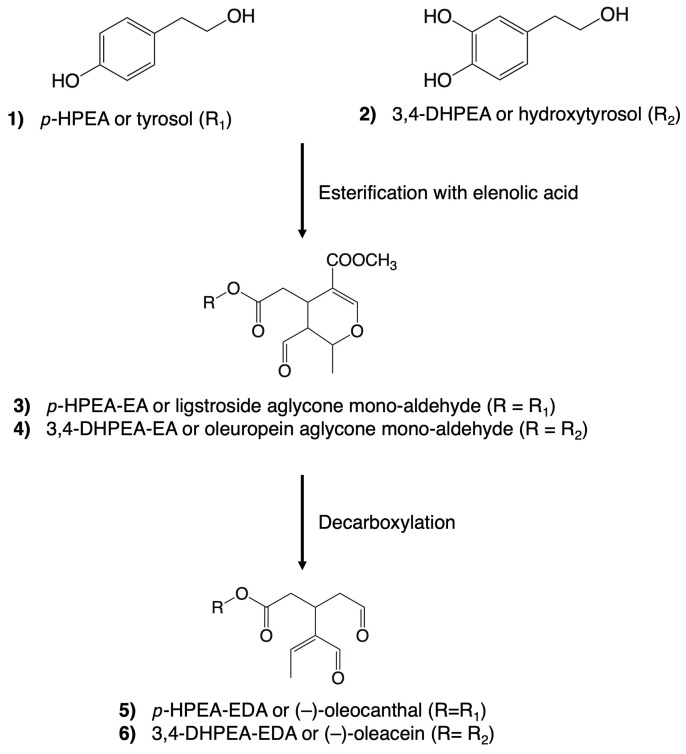
Chemical structures of some of the most abundant phenolic compounds in virgin olive oil. A simplified version of the natural derivatization process is illustrated.

**Figure 3 nutrients-16-01283-f003:**
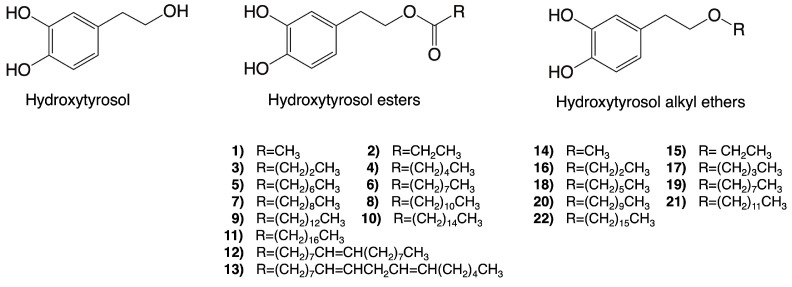
Molecular structures of some of the synthetic derivatives of hydroxytyrosol. Structures of hydroxytyrosol, hydroxytyrosol esters (**1**–**13**), and hydroxytyrosl alkyl ethers (**14**–**22**) are shown.

**Table 1 nutrients-16-01283-t001:** Virgin olive oil composition.

Component	Concentration
**Lipids**	
*Acylglycerols*	(%)
Triacylglycerols	Up to 99.0
Diacylglycerols	1.0–2.8
Monoacylglycerols	0.3
*Fatty acids in acylglycerols*	(%)
Lauric C12:0	Not detectable
Myristic C14:0	<0.1
Palmitic C16:0	7.5–20.0
Palmitoleic C16:1	0.3–3.5
Heptadecanoic C17:0	<0.5
Heptadecenoic C17:1	<0.6
Stearic C18:0	0.5–5.0
Oleic C18:1	55.0–83.0
Linoleic C18:2	3.5–21.0
Linolenic C18:3	≤1.0
Arachidic C20:0	0.3–0.8
Eicosenoic C20:1	≤0.4
Docosanoic C22:0	0.09–0.12
Lignoceric C24:0	≤0.2
Total MUFA content	65.2–80.8
Total PUFA content	7.0–15.5
*Tocopherols*	(mg/kg)
α-Tocopherol	10.2–208.0
β-Tocopherol	0.8–10.0
γ-Tocopherol	0.7–20.0
*Pigments*	(mg/kg)
Total chlorophylls	0.2–62.0
Pheophytin-a	0.1–0.5
Total carotenoids	0.5–31.5
β-Carotene	0.2–0.7
Lutein	0.7–3.6
*Other lipids*	
Squalene (mg/kg)	200.0–8260.0
Triterpene dialcohols (% of total sterols)	0.9–2.8
β-sitosterol (mg/kg)	530.2–2638.6
**Other Compounds**	
Total phenolic compounds (mg/kg)	50.0–940.0

Data retrieved from [29,31,32,33].

**Table 2 nutrients-16-01283-t002:** Phenolic compounds in virgin olive oils.

Family	Compound
Phenyl ethyl alcohols	Tyrosol (*p*-hydroxyphenyl ethanol) or *p*-HPEA
	Hydroxytyrosol (3,4-dihydroxyphenyl ethanol) or 3,4-DHPEA
Secoiridoid aglycons	Oleuropein aglycon or 3,4-DHPEA-EA
	Ligstroside aglycon or *p*-HPEA-EA
	Aldehydic form of oleuropein aglycon
	Aldehydic form of ligstroside aglycon
Dialdehydic forms of secoiridoids	(-)–Oleocanthal (decarboxymethyl ligstroside aglycon) or *p*-HPEA-EDA
	(-)–Oleacein (decarboxymethyl oleuropein aglycon) or 3,4-DHPEA-EDA
Benzoic and derivative acids	3-Hydroxybenzoic acid
	*p*-Hydroxybenzoic acid
	3,4-Dihydroxybenzoic acid
	Gentisic acid
	Vanillic acid
	Gallic acid
	Syringic acid
Cinnamic acids and derivatives	*o*-Coumaric acid
	*p*-Coumaric acid
	Caffeic acid
	Ferulic acid
	Sinapinic acid
Flavonols	(+)-Taxifolin
Flavones	Apigenin
	Luteolin
Lignans	(+)-Pinoresinol
	(+)-1-Acetoxypinoresinol
	(+)-1-Hydroxypinoresinol
Hydroxyisochromans	1-Phenyl-6,7-dihydroxyisochroman
	1-(3′-Methoxy-4′-hydroxy)phenyl-6,7-dihydroxy-isochroman
Others	Verbascoside

Data retrieved from [28,29,31,39].

**Table 3 nutrients-16-01283-t003:** Molecular mechanisms in antioxidant effects of key phenolic compounds from VOO.

Phenolic Compound	Molecular Mechanism	References
HT	Free radical (O_2_^∙−^, OH^∙^, ONOOH) scavenger	[44,45,46]
	Iron chelating	[47]
	Promotion of antioxidant systems (CAT, SOD, GPx, Nrf2)	[48,49,50,51,52,53]
(-)–Oleocanthal	Free radical (O_2_^∙−^, H_2_O_2_, HOCl) scavenger	[54]
(-)–Oleacein	Free radical (O_2_^∙−^, H_2_O_2_, HOCl) scavenger	[55,56]
HT esters	Free radical (DPPH^∙^…) scavenger	[57,58,59]
	Activation of Nrf2	[60]
	Activation of AKT and ERK1/2 signaling	[60]
HT alkyl ethers	Increase in GSH production	[61,62]
	Reduction in lipid peroxidation	[63]
	Promotion of antioxidant systems (GPx, GR)	[62]

HT: hydroxytyrosol; CAT: catalase; SOD: superoxide dismutase; GPx: glutathione peroxidase; AKT: protein kinase B; ERK1/2: extracellular signal-regulated kinase; GSH: reduced glutathione; GR: glutathione reductase.

**Table 4 nutrients-16-01283-t004:** Anti-cancer effects of key phenolic compounds from VOO.

Compound	Biological Effect	Molecular Mechanism	References
HT	Reduction in tumor cell proliferation	Inhibition of ERK1/2 signaling	[77,78]
		Inhibition of EGFR signaling	[79]
		Modulation of Wnt pathway	[80]
		Cell cycle arrest	[77,80,81,82,83,84,85,86,87,88]
	Reduction in tumor cell survival	Inhibition of the pro-survival AKT pathway	[88]
		Modulation of Wnt pathway	[80]
		Induction of apoptosis via caspase activation and mitochondrial pathway	[80,83,85,86,87]
	Cytotoxic	Specific pro-oxidant action	[89,90,91]
(-)–Oleocanthal	Reduction in tumor cell proliferation	Inhibition of ERK1/2 signaling	[92,93,94]
		Inhibition of HGF signaling	[95,96]
		Inhibition of STAT3 signaling	[97]
		Cell cycle arrest	[95]
	Reduction in tumor cell survival	Inhibition of the pro-survival AKT pathway	[93,94]
		Induction of apoptosis via caspase activation and mitochondrial pathway	[92,94]
(-)–Oleacein	Reduction in tumor cell proliferation and survival	Inhibition of AKT and ERK1/2 signaling	[93]
HT esters	Reduction in tumor cell proliferation and survival	Inhibition of AKT and ERK1/2 signaling	[58,98]
		Cell cycle arrest	[99]
		Induction of apoptosis via caspase activation and mitochondrial pathway	[100]
	Reduction in cell migration	-	[98]
HT alkyl ethers	Reduction in tumor cell proliferation	Inhibition of AKT and ERK1/2 signaling	[98]
	Cytotoxic	Induction of oxidative stress	[101]
	Reduction in cell migration	-	[98]

EGFR: epidermal growth factor receptor; HGF: hepatocyte growth factor.

**Table 5 nutrients-16-01283-t005:** Modulation of angiogenesis-related mechanisms by key phenolic compounds from VOO.

Compound	Biological Effect	Molecular Mechanism	References
HT	Reduction in pro-angiogenic signaling	Inhibition of COX-2	[107]
		Inhibition of the pro-survival AKT pathway	[88,108]
		Inhibition of VEGFR signaling	[108]
		Inhibition of NF-κB signaling	[88]
	Decrease in extracellular remodeling capacity	Reduction in MMP expression	[71,107,109]
HT alkyl ethers	Cytotoxic	Induction of apoptosis	[110]
	Reduction in endothelial cell migration and tube formation	-	[110]
(-)–Oleocanthal	Reduction in endothelial cell migration, invasion, and tube formation	Inhibition of AKT and ERK1/2 signaling	[111]
		Inhibition of STAT3 signaling	[97]
(-)–Oleacein	Reduction in endothelial cell migration, invasion, and tube formation	Inhibition of AKT and ERK1/2 signaling	[111]

COX-2: cyclooxygenase 2; VEGFR: vascular endothelial growth factor receptor; NF-κB: nuclear factor-kappa B; MMP: matrix metalloproteinase.

**Table 6 nutrients-16-01283-t006:** Anti-atherosclerotic effects of key phenolic compounds from VOO.

Phenolic Compound	Biological Effect	Molecular Mechanism	References
HT	Protection versus LDL oxidation	Antioxidant mechanisms previously described	[45,49,50]
	Hypolipidemic	-	[49,50]
	Anti-inflammatory	Inhibition of NF-κB signaling	[51,72,116,117]
		Inhibition of COX-2	[116,118]
		Inhibition of iNOS and NO production	[44,116,117,118,119]
		Reduction in proinflammatory cytokines’ (i.e., TNF-α) and chemokines’ (i.e., CCL2, CXCL10) expression	[51,117]
		Reduction in adhesion molecules’ (i.e., ICAM-1, VCAM-1, E-selectin) expression.	[72,120,121,122]
(-)–Oleocanthal	Anti-inflammatory	Inhibition of NF-κB signaling	[55,123]
		Inhibition of COX-2	[55,123,124]
		Inhibition of iNOS and NO production	[123,125]
(-)–Oleacein	Anti-inflammatory	Inhibition of NF-κB signaling	[55,56,126]
		Inhibition of COX-2	[55,126,127]
		Inhibition of iNOS and NO production	[127]
HT esters	Anti-inflammatory	Inhibition of COX-2	[127,128]
		Inhibition of iNOS and NO production	[127,128]
		Reduction in proinflammatory cytokines (i.e., TNF-α, IL-1β, IL-6)	[127,128,129]
		Decrease in prostaglandins production	[127,128]
HT alkyl ethers	Anti-inflammatory	Inhibition of NF-κB signaling	[130]
		Inhibition of COX-2	[131]
		Reduction in IL-1β production	[63,131]

iNOS: inducible nitric oxide synthase; NO: nitric oxide; CCL2: C-C motif chemokine ligand 2; CXCL10: C-X-C motif chemokine ligand 10; ICAM-1: intercellular adhesion molecule 1; VCAM-1: vascular adhesion molecule 1; TNF-α: tumor necrosis factor-alpha; IL-1β: interleukin 1-beta; IL-6: interleukin 6.

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
