# Peer review of "Anti-Cancer, Anti-Angiogenic, and Anti-Atherogenic Potential of Key Phenolic Compounds from Virgin Olive Oil"

_nutrients, 2024, doi:10.3390/nu16091283_

Round 1
Reviewer 1 Report
Comments and Suggestions for Authors
I appreciate the opportunity to evaluate this manuscript delving into the biological activities of minor components found in olive oil. The focal point, as indicated by the title, revolves around angiopreventive, oncopreventive, and chemopreventive properties. However, several critical issues need addressing before the review is deemed ready for publication.
Major Concerns:
-
Misleading Title: The title may mislead readers due to the limited information on angioprevention, oncoprevention, and chemoprevention, with the focus predominantly on antioxidant activities of phenolic compounds. To better align the title with the content, I recommend to either modify the title to accurately reflect the review's scope or, more importantly, enhance the review by including information on the stated preventive properties of all minor components in olive oil. This broader perspective would indeed constitute the novelty of the review, especially considering the extensive coverage of phenolic compounds' antioxidant and anti-inflammatory activities in existing literature. An alternative approach could be to explore the research conducted on synthetic derivatives of hydroxytyrosol.
-
Use of "Extra" in Title and Text: The term "extra" in reference to virgin olive oil's health effects is misleading, as it pertains solely to organoleptic qualities and free fatty acid content, not to the presence of nutrients and bioactive compounds. The text should explicitly clarify this distinction, and the term "extra" should be removed from both the title and the body of the text.
-
Clarification on Oil Extraction Process: Virgin Olive Oil (VOO) and Extra Virgin Olive Oil (EVOO) are extracted by physical means only but the manusript stated that some physical procedures are prohibited. It is essential to correct this by acknowledging that physical means can include processes such as centrifugation, decanting, and filtration, in accordance with legislation.
-
Inclusion of Phenolic Compound Table: To enhance clarity and accessibility, a table summarizing the various phenolic compounds present in VOO and EVOO should be incorporated.
-
In-depth Discussion of Unsaponifiable Fraction: Given that the unsaponifiable fraction of olive oils, devoid of phenolics, is attributed to numerous health benefits, the review should extensively discuss this aspect in alignment with the title.
-
Insufficient Randomized Controlled Trials (RCTs): There is a notable lack of discussion on randomized controlled trials and their meta-analyses concerning the antioxidant activities of VOO phenolics. This is a crucial gap that warrants thorough review and discussion.
- Strcuture: The manuscript is not well structurated. For instance, point 3 deals with Bioactive Phenolic Compounds in Extra Virgin Olive Oil, Point 3.1. Tyrosol, Hydroxytyrosol, and Derivatives and 3.2. –(-)Oleocanthal and –(-)Oleacein but 3.3. is about authophagy and is differntly formatted. Moreover, the concluding remarks are also numbered as 3.
Minor Issues:
-
Figure 1 Reference: A reference should be added to Figure 1 for clarity and academic rigor.
-
Typo in Olive Tree Systematic Name: Correct the typo in the systematic name of the olive tree for accuracy.
-
Clarity on "High-Quality Oils": Specify which high-quality oils are being referred to in line 94 for precision.
-
Palimitic Acid Content Clarification: The sentence on lines 101-102.6 should not hide that the palmitic acid content is commonly higher than that of linoleic acid in most olive oils.
-
Abbreviation Check: Verify and ensure consistency in the use of abbreviations throughout the manuscript.
-
Table 2 and Table 4 Enhancements: Include observed effects of tyrosol in Table 2, and in both Table 2 and Table 4, incorporate details about experimental models and sample sizes for a comprehensive understanding.
Reviewer 2 Report
Comments and Suggestions for Authors
I reviewed the manuscript titled “Angiopreventive, Oncopreventive and Chemopreventive Bioactive Compounds of Extra Virgin Olive Oil”. This review has quality and can be considered after addressing below suggestions.
Title can be revised as Angiopreventive, Oncopreventive and Chemopreventive potential of Bioactive Compounds extracted from Extra Virgin Olive Oil”
Abstract
Authors must provide review findings, conclusions, and recommendations in the abstract.
Keywords can be revised as these are already in title
Authors should provide graphical abstract
Introduction
Introduction. Mediterranean Diet: More Than a Dietary Choice. This can be revised as “ Mediterranean Diet: More Than a Dietary Choice”
There are no clear review objectives in the introduction
Figure 1 should be revised for high pixel
Table 1. * mark is not cited in Table
Figure 2: kindly provide copyright permission. Adapted from [38] should be revised as Adapted with permission (copyright holder, location) from xxxx et al. [38]. Follow similar format for Figure 3
Authors must specify the application of oil bioactive compounds as a separate sub-section for each activity. For example,
Bioactive potential of Extra Virgin Olive Oil
1. Angiopreventive: discuss all Angiopreventive activity with examples and relevant literature
2. Oncopreventive: discuss all Oncopreventive activity with examples and relevant literature
3. Chemopreventive: discuss all Chemopreventive activity with examples and relevant literature
4. Authors merge above sections, leading to confusion
A TABLe containing above activities, bioactive compounds concentration, and mechanism to act as Angiopreventive, Oncopreventivem Chemopreventive should be provided in a Table
References should be cross-checked and adjusted according to journal guidelines.
Round 2
Reviewer 1 Report
Comments and Suggestions for Authors
All questions raised by the reviewer has been answered and corrections have been made.
Reviewer 2 Report
Comments and Suggestions for Authors
The quality of the manuscript is now improved. In my opinion, this version can be accepted for possible publication consideration
Comments on the Quality of English LanguageThe quality of the manuscript is now improved. In my opinion, this version can be accepted for possible publication consideration